# Technical Solution for Monitoring Climatically Active Gases Using the Turbulent Pulsation Method

**DOI:** 10.3390/s23208645

**Published:** 2023-10-23

**Authors:** Ekaterina Kulakova, Elena Muravyova

**Affiliations:** Department of Automated Technological and Information Systems, Institute of Chemical Technology and Engineering, Ufa State Petroleum Technological University, Sterlitamak 453103, Russia; muraveva_ea@mail.ru

**Keywords:** monitoring, Arduino, multicomponent control system, atmospheric air

## Abstract

This article introduces a technical solution for investigating the movement of gases in the atmosphere through the turbulent pulsation method. A comprehensive control system was developed to measure and record the concentrations of carbon dioxide and methane, temperature, humidity, atmospheric air pressure, wind direction, and speed in the vertical plane. The selection and validation of sensor types and brands for each parameter, along with the system for data collection, registration, and device monitoring, were meticulously executed. The AHT21 + ENS160 sensor was chosen for temperature measurement, the BME680 was identified as the optimal sensor for humidity and atmospheric pressure control, Eu-M-CH4-OD was designated for methane gas analysis, and CM1107N for carbon dioxide measurements. Wind direction and speed are best measured with the SM5386V anemometer. The control system utilizes the Arduino controller, and software was developed for the multicomponent gas analyzer.

## 1. Introduction

Currently, the method of turbulent pulsations is extensively employed in studying gas movement within the atmosphere [1,2,3]. The method centers on quantifying the number of molecules that experience vertical movement over a specific timeframe. The calculation of turbulent flow necessitates knowledge of vertical airstream direction, substance concentration, and density. This method enables the determination of ascending and descending gas flows, heat transfer, and moisture movement. The turbulent gas flow in the atmosphere is determined as the product of average air density and the covariance of vertical wind speed and admixture ratio. Moreover, this approach describes the temperature and velocity pulsations in turbulent atmospheric airflow as follows:F ≈ ρ_a_·w·s,(1)
where F—turbulent flow, $\frac{g}{m^{6}}\cdot \mathrm{degree}$;

ρ_a_—air density, kg/m^3^;

w—vertical wind speed covariance, $\mathrm{degree}\cdot \frac{\mathrm{mg}}{m^{3}}$;

s—ratio of gas density to air density.

When using the turbulent pulsation method, it is assumed that changes in air density are negligible, and there is no divergence and convergence of the vertical airstream.

Researchers globally widely employ the pulsation method. For instance, the carbon cycle of the Middle Tien Shan pasture ecosystem has been analyzed, revealing fluctuations in carbon balance during daylight hours and days. Gross primary productivity and ecosystem respiration have been determined, along with months of carbon dioxide sources and absorbers. Notably, soil respiration is significantly influenced by temperature [4]. Similarly, researchers at the University of Exeter employed the eddy covariance method to calculate the carbon balance in the Chihuahuan Desert (North America) [5]. Eddy covariance was also applied to study carbon dioxide flux through the air–sea interface, establishing gas flow resistance coefficients based on roughness [6]. 

The foundation of the turbulent pulsation method lies in its instrumentation, allowing requisite parameter recording [7,8]. To accomplish this, gas analyzers, temperature, humidity, and atmospheric pressure sensors, as well as horizontal and vertical wind direction and speed sensors, are essential. Currently available instruments are discrete control devices—gas analyzers, weather stations, and anemometers—often single-component, determining only one parameter. For comprehensive scientific research, synchronous measurement and registration of parameters demand a comprehensive solution [9].

A predominant solution on the market is the vortex covariance system by LiCor (USA) [10,11,12,13], providing gas analyzers for CO_2_, H_2_O, and CH_4_ measurements [14,15]. LiCor offers open- and closed-type carbon dioxide gas analyzers, such as LI-7500DS and LI-7200RS, respectively. LI-7200RS is a closed-type CO₂/H₂O gas analyzer. Table 1 outlines a comparative analysis of LiCor open- and closed-type gas analyzers.

These devices are currently used across the Eurasian continent, as well as in North and South America, and within research laboratories such as the Natural Resource Ecology Laboratory at Colorado State University in Fort Collins, Colorado, the Pacific Northwest National Laboratory, ICOS Atmospheric Thematic Centre, Laboratoire des Sciences du Climat et de l’Environnement [16,17], as well as the United Kingdom’s National Physical Laboratory.

Depending on the customer’s requirements, the systems are equipped with sensors for temperature, humidity, rain, illumination, and a radiometer [18]. The systemic approach implemented by eddy covariance enables the comprehensive study of gas exchange in a given area. Nevertheless, a significant challenge associated with this solution is its high cost, which stands as a crucial limiting factor in conducting scientific experiments.

Picarro introduced the G2508 analyzer to the market, offering simultaneous and accurate measurement of nitrous oxide (N_2_O), methane (CH_4_), carbon dioxide (CO_2_), ammonia (NH_3_), and water vapor (H_2_O). Its specifications are detailed in Table 2.

The measurement method is based on precise ring spectroscopy of the resonator. This highly sensitive optical spectroscopic method facilitates the measurement of the absolute optical absorption in samples that scatter and absorb light. Widely employed for analyzing gas samples that absorb light at specific wavelengths, it enables the determination of mole fractions down to parts per trillion. Commonly referred to as cavity ring laser absorption spectroscopy (CRLAS) [19], this method has found popularity among Latvian scientists engaged in researching carbon exchange in agricultural land [20,21,22,23,24].

The G2508 measures and records atmospheric gases. To implement the turbulent pulsation method, supplementary instruments for measuring meteorological parameters are necessary.

The equipment market for conducting pulsation studies primarily comprises control devices designed to measure and record climatically active gases in most cases. These devices lack the capability to expand the roster of controlled gases; they solely monitor the concentration of a specific gas. Expanding research objectives and measuring other substances necessitate acquiring new control devices. Consequently, this approach hampers the swiftness of responding to novel research challenges.

Thus, the development of flexible new technical capacities for research becomes imperative. One plausible approach involves employing separate devices and integrating them into a unified monitoring system.

The aim of the study is to devise a technical solution for conducting comprehensive investigations into the motion of climatically active gases using the turbulent pulsation method.

The study has the following objectives:Development of a prototype for a multicomponent gas analyzer;Selection of discrete and analog input modules to receive unified signals from sensors;Design of a data transmission system in a multicomponent gas analyzer.

## 2. Materials and Methods

This work employs the fundamentals of constructing an automated process control system (APCS) as its research methodology [25,26,27]. APCS revolves around the interconnection and coordination of control devices and systems (storage, processing, operations) for greenhouse gases.

To construct a gas analyzer, sensors are indispensable. The selection of sensors, essential for a multicomponent gas analyzer, was performed by examining the array of sensors available on the global market.

Temperature sensors, namely CJMCU-811, GY-SGP30, and AHT21 + ENS160, were considered. These sensors exhibit distinct characteristics, with the choice hinging on specific requirements and objectives. A comparative analysis of temperature measurement sensors was conducted (Table 3).

CJMCU-811 and GY-SGP30 are sensors designed for measuring air quality, with temperature measurement being an additional function. However, their primary purpose is not temperature measurement. The CJMCU-811 sensor measures CO_2_ and VOC (volatile organic compounds), while the GY-SGP30 sensor measures CO_2_ and TVOC (total volatile organic compounds). These sensors have found utility in indoor air quality monitoring.

The AHT21 + ENS160 sensor was selected for measuring atmospheric air temperature. The AHT21 and ENS160 sensors boast a broader operating temperature range, elevated measurement precision, and reduced noise levels. Furthermore, these sensors consume less power, enhancing their efficiency. They also offer support for the I2C interface, facilitating their integration with the Arduino platform.

To measure humidity, temperature, and air pressure, sensors like BMP280, LPS22HB, and BME680 are employed. Each of these sensors has distinct characteristics and applications. A comparative analysis of humidity and pressure measurement sensors was undertaken (Table 4).

The BME680 sensor brings forth several advantages:Additional Dimension. The BME680 introduces an additional dimension—air quality assessment. This sensor gauges gas concentrations such as nitrogen oxides, ammonia, ethanol, and carbon dioxide to evaluate air quality. This capability is absent in BMP280 and LPS22HB.Enhanced Precision. The BME680 exhibits superior humidity, pressure, and temperature measurement accuracy compared to BMP280 and LPS22HB. It also delivers enhanced measurement stability with reduced noise in output signals.Wide Measurement Range. The BME680 covers a wide measurement range—from 0 to 100% for humidity, from 300 to 1100 hPa for pressure, and from −40 to +85 °C for temperature. This wide range allows the sensor to be used for a variety of purposes.Multiple Interfaces. The BME680 supports a variety of interfaces, including I2C, SPI, and UART, rendering it compatible with an array of microcontrollers and devices.User-Friendly. Compact and lightweight, BME680 facilitates easy integration into diverse systems. Additionally, it consumes less power, making it suitable for energy-constrained mobile devices.

Table 5 outlines some CO_2_ sensors that can be employed for atmospheric CO_2_ detection, along with their characteristics.

Most sensors available on the market are optical sensors. Depending on the intended application, sensors with an error range of 50 ppm + 3% to 1% can be procured. The CM1107N is a favorable choice for carbon dioxide measurement in terms of its price-to-performance ratio. It operates effectively across a wide temperature range, providing sufficient accuracy and response time for measurements.

Sensors for measuring methane in atmospheric air were also considered: MSH-P/HCP, Eu-M-CH4-OD, F3-042107-05000, and Gasboard-2500. A comparative analysis of these methane sensors is presented in Table 6.

As depicted in Table 6, Russia also produces methane sensors. These sensors exhibit low measurement error and rapid signal response times. Eu-M-CH4-OD was selected as the methane sensor due to its extended operating temperature range and narrow gas concentration measurement range. Given the marginal global fluctuations in atmospheric methane, approximately 110 ppm, precision is paramount in the measurement instrument.

Anemometer sensors, namely SM5386V, RK100-02, and BGT-FSI-RS485, possess distinct specifications, with the ideal choice depending on specific requirements and applications. A comparative analysis of these anemometers was conducted (Table 7).

Both SM5386V and RK100-02 are analog air speed sensors, which gauge air speed in meters per second and yield an analog voltage signal contingent on the air speed. In contrast, the BGT-FSI-RS485 sensor quantifies air speed in m/s and generates a digital signal utilizing the RS485 interface.

Among these, the SM5386V anemometer was preferred for the following reasons:Extensive Measurement Range. With a measurement range from 0.5 m/s to 30 m/s, the SM5386V accommodates a wide spectrum of conditions.High Precision. The SM5386V ensures accurate air velocity measurement, down to 0.1 m/s, making it suitable for scientific research.User-Friendly. Compact and easily interfaced with microcontrollers through analog inputs, the SM5386V stands out, while RK100-02 and BGT-FSI-RS485 require additional hardware for signal processing.Reliability. The SM5386V boasts minimal noise and an extended service life, enhancing its reliability. In contrast, RK100-02 and BGT-FSI-RS485 may encounter noise-related issues and signal processing requirements.

The Arduino Mega 2560 [28] was chosen as the controller due to its user-friendly nature. Minimal skills in working with digital automation devices are required to work with projects based on this controller. The Arduino IDE software shell, used for creating sketch programs (firmware), uploading code, and monitoring data exchange, is constantly updated. Notably, the Arduino IDE features high-quality add-ons and extensions, including those that support 32-bit microcontroller programming. The Arduino boards do not necessitate specialized loader-debugger or programming tools; the main work was executed through the Arduino platform.

Furthermore, the device mandates a power supply, a power protection module, and a data communication module for operation. The SIM800L GSM data transmission module was employed.

The multi-component measuring tool consists of the Arduino controller, AHT21 + ENS160 temperature sensor, BME680 air pressure and humidity sensor, Eu-M-CH4-OD methane sensor, CM1107N carbon dioxide sensor, SM5386V anemometer, SIM800L GSM communication module, DC power supply, and power board protection module.

## 3. Results

A prototype of a device for monitoring the parameters required for research using the turbulent pulsation method was developed. It performs the following functions:Measurement and recording of greenhouse gas concentrations;Measurement and registration of atmospheric air temperature;Measurement and registration of wind speed and direction in the vertical plane;Measurement and registration of atmospheric air humidity;Measurement and registration of atmospheric air pressure.

The developed system consists of a personal computer with an installed automated control system, 1—controller (for example, Arduino), 2—relay with connected sensors, 3—power supply, 4—sampling pump, and 5—communication module (for example, a GSM modem based on the SIM800L chip) (Figure 1). The complex of sensitive elements includes the following sensors: 2_1_—sensor for measuring the concentration of carbon dioxide, 2_2_—sensor for measuring the concentration of methane, 2_3_—sensor for measuring the humidity of atmospheric air, 2_4_—sensor for measuring the temperature of atmospheric air, 2_5_—sensor for measuring the atmospheric air pressure, 2_6_—sensor for measuring the speed wind in the horizontal direction, 2_7_—sensor for measuring the wind direction in the horizontal plane, 2_8_—sensor for measuring the direction of vertical air movement, and 2_9_—sensor for measuring the speed of vertical air movement. An external memory module (1_1_) (SD card module) is connected to controller 1.

The automated system 1 collects, registers, and displays data recorded by the sensors 2_1_, 2_2_, …, 2_9_.

The output voltage of power supply 3 must match the required voltage of the controller and relay 2 (most devices require 7–12 V).

Data transfer from modules 2_1_, 2_2_, …, 2_9_ to relay 2 occurs via I2C, UART, SPI, and other Modbus protocol interfaces.

Data transfer from controller 1 to PC occurs through the RS-232 interface.

The principle of operation of the device is as follows: When pump 4 is operational, signals from sensors 2_1_, 2_2_, …, 2_9_ are fed to relay 2, which collects information from all the control devices. Experimental data is then transmitted from relay 2 to controller 1 via the Modbus protocol. The data are recorded in registers and sent to a PC via the RS-232 interface. On the PC, in the automated control system (SCADA-system), the current values from the control devices are displayed and recorded in the database. Measurement data of sensors 2_1_, 2_2_, …, 2_9_ are stored on an external storage device (e.g., an SD card) using module 1_1_. Data arrays are recorded in table formats. File format: *.txt, *.csv, *.xls, etc. Communication module 5 transfers information according to the Groupe Special Mobile digital mobile cellular communication standard to external information storage (e.g., cloud databases).

### 3.1. Connecting Sensors of the Device Prototype to the Control Board

To connect the BME680 temperature, humidity, and pressure sensor, use the following steps: The BME680 sensor has four pins: VCC, GND, SLA, and SDA, which serve as the analog signal output (Figure 2). This sensor requires a 5 V DC voltage source for power. Connect the positive voltage to “5 V” and the ground to the “GND” input on the board. SLA and SDA should be connected to analog inputs.

The ENS160 + AHT21 sensor is connected to the pins, the SDA and SCL contacts (Figure 3). The module board includes a supply voltage stabilizer and a level converter for the I2C bus. VCC can be connected to either 5 or 3.3 volt logic, while the ground is connected to the GND input on the board.

To connect the SIM800L GSM communication module, a separate 4.0 V supply voltage regulator is required (Figure 4). A resistive voltage divider is used to match the logical levels of the signals sent to the Arduino controller.

Boosting DC–DC Converter XL6009: The XL6009 module has input and output voltage terminals. LEDs are placed on the board, indicating its operating status.

Since this is a boost converter, the supply potential at the input of the module will always be less than at the output. Therefore, it is quite possible to obtain a voltage at the output greater than at the input using XL6009. To increase its voltage (current), it is necessary to rotate the corresponding knob of the potentiometers located on the board clockwise. To decrease, turn it in the other direction. This allows for the adjustment of the required values of the output parameters.

Out of all 12 pins, only 4 are needed: VCC, RXD, TXD, and GND.

To ensure the stable operation of the SIM800L module, a power supply with an output voltage of 3.4–4.4 V and a maximum operating current of 2 A is required. Thus, connect the VCC and GND contacts to the plus and minus terminals of the 5 V power supply, and then connect RXD and TXD to the pins.

To create an equal-arm voltage divider, use a pair of resistors of the same rating within the range of 1–10 KΩ. This configuration will provide a voltage of 2.5 V, which fits within the required range of 2.1–3.1 V.

It is not possible to power the module from the Arduino. The module can draw up to 2A, and the Arduino is not capable of supplying such a current. Attempting to power the module from the Arduino or the USB port of the computer could result in damage to one or both components. This is particularly relevant in the active operation mode when transferring data to the database, as the current consumption increases significantly.

Connecting the wind speed sensor and the 18650 Li-ion battery protection board. First, short wires with a cross section of 0.5–0.75 mm should be prepared. It is necessary to solder them to the board pads for added reliability and to reduce the risk of accidental short-circuits of the battery leads (Figure 5). First, soldering the extreme terminals of the assembly is needed.

Next, the connection of the middle output is carried out, where the cross-section of the wire is not critical as it does not carry a large current. Finally, all the components should be connected together using the appropriate wires and terminal blocks.

In summary, the VIN and GND contacts are connected to the plus and minus of the p+ and p- protection board and to the plus and minus of the 5 V power supply. Power is supplied to the device and connected to the analog input.

To connect the Eu-M-CH4-OD methane sensor to the Arduino board, it is necessary to:Connect the +V pin of the sensor to the 5 V pin on the Arduino Mega 2560.Connect the GND pin of the sensor to the GND pin on the Arduino Mega 2560.Connect the AO analog input pin of the sensor to the AO pin on the Arduino Mega 2560.Connect the Rx pin of the sensor to the Rx pin on the Arduino Mega 2560.Connect the Tx pin of the sensor to the Tx pin on the Arduino Mega 2560.

The wiring diagram for connecting the Eu-M-CH4-OD methane sensor is illustrated in Figure 6.

To connect the CM1107N carbon dioxide sensor to the Arduino board, it is necessary to:Connect the VCC pin of the sensor to the 5 V pin on the Arduino Mega 2560.Connect the GND pin of the sensor to the GND pin on the Arduino Mega 2560.Connect the SDA pin of the sensor to the SDA pin on the Arduino Mega 2560.Connect the SCL pin of the sensor to the SCL pin on the Arduino Mega 2560.

The wiring diagram for connecting the CM1107N carbon dioxide sensor is illustrated in Figure 7.

The wiring diagram of the device prototype was developed and is illustrated in Figure 8.

The measurement system (Figure 8) consists of a controller 1, sensors for climatically active gases 2 (carbon dioxide, methane), sensors for meteorological parameters 3 (temperature, humidity, atmospheric air pressure), anemometer 4, remote transmission module 5, power balancing module 6, and electrical energy sources 7. 

The developed control device is illustrated in Figure 9.

### 3.2. Software and Hardware Implementation of the Device Prototype

The program for managing the greenhouse gas control process was created using the Arduino IDE. The variables for the controlled and adjustable parameters of the technological process are defined in the variable editor window (Figure 10).

### 3.3. Testing

Field tests of the developed control device were conducted in urbanized areas with low-growing mixed vegetation at a height of 110 mm from the ground level during the period from 4 to 6 in the afternoon. The anemometer was positioned to register a 0° angle when air moved vertically upwards and 180° when moving vertically downwards.

Time series data on the concentration of carbon dioxide, methane, temperature, humidity, atmospheric air pressure, wind direction, and speed in the vertical direction were collected (Figure 11).

According to the data obtained, the turbulent flow of carbon dioxide, calculated by formula 1, is −63.478 $\frac{g}{m^{6}}\cdot \mathrm{degree}$, and methane measures at −0.098 $\frac{g}{m^{6}}\cdot \mathrm{degree}.$ Thus, during the period of operation of the control device, there are prevailing downward flows of CO_2_ and CH_4_, with CO_2_ exhibiting greater intensity.

## 4. Conclusions

This paper proposes a technical solution for measuring and recording the parameters necessary for conducting scientific studies on the movement of gas flows in atmospheric air using the method of turbulent pulsations. This solution consists of a complex device, including a controller and connected measuring elements. The system can be easily upgraded to accommodate additional sensors, enabling rapid adaptation to changes in the goals and objectives of scientific research, which provides a competitive advantage over existing market solutions.

The article also discusses organizational and technical considerations during the installation of automation systems. It justifies the choice of technical and software components for automation, including the selection of the Arduino controller and sensors such as AHT21 + ENS160, BME680, Eu-M-CH4-OD, CM1107N, and SM5386V. Furthermore, it explains the rationale behind the selection of mounting and auxiliary automation and electrical equipment. The programming of the Arduino Mega 2560 controller is performed using the Arduino IDE software, resulting in the successful creation of a greenhouse gas control system with real-time technological parameters.

Field tests of the developed measuring device were conducted, yielding valuable data on the turbulent flow of carbon dioxide and methane.

The developed measuring tool is technically straightforward to implement and user-friendly. It offers cost-effectiveness, and its characteristics can be easily customized by adding new sensors as needed.

## Figures and Tables

**Figure 1 sensors-23-08645-f001:**
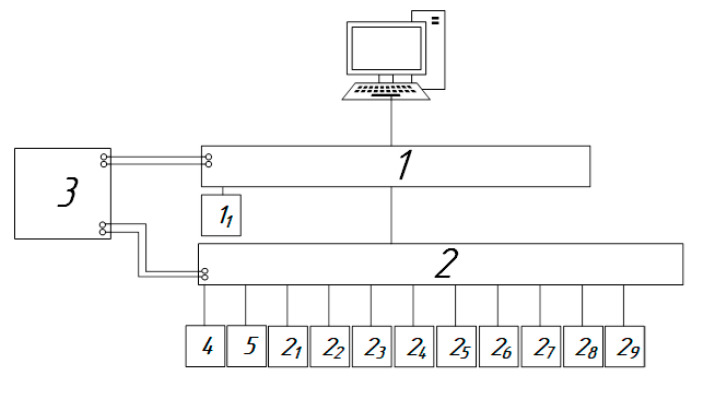
Structural diagram of the device.

**Figure 2 sensors-23-08645-f002:**
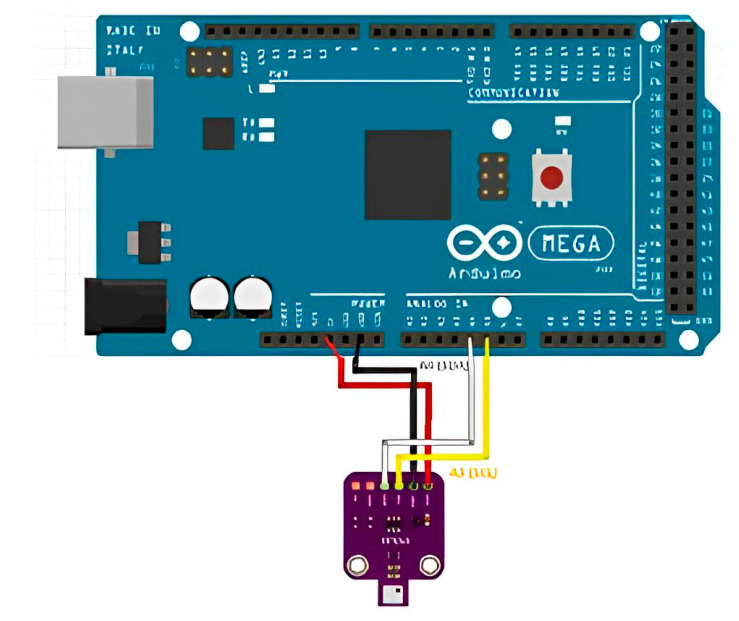
BME680 sensor connection diagram.

**Figure 3 sensors-23-08645-f003:**
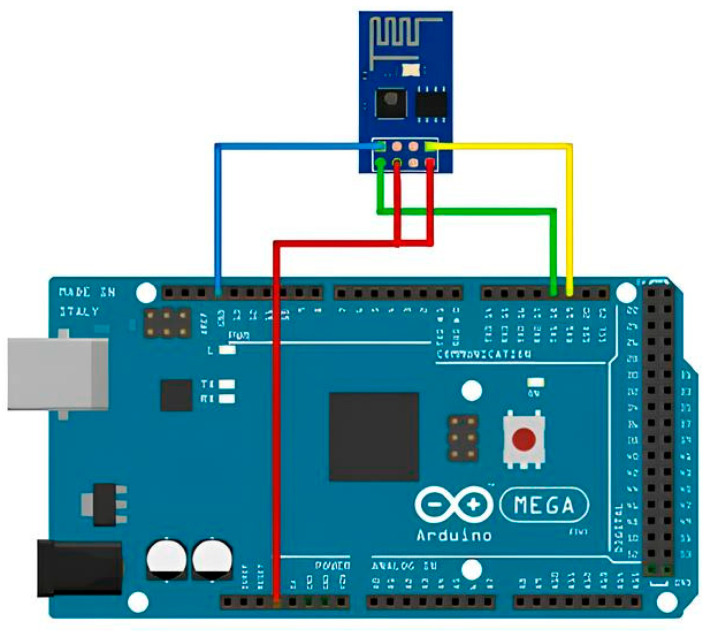
ENS160 + AHT21 sensor connection diagram.

**Figure 4 sensors-23-08645-f004:**
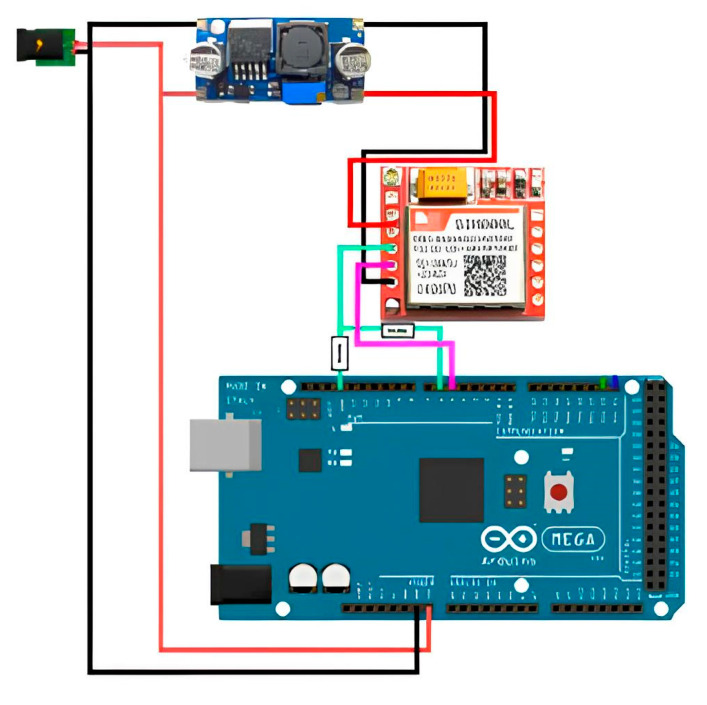
Wiring diagram of SIM800L GSM communication module and XL4015 DC-DC converter.

**Figure 5 sensors-23-08645-f005:**
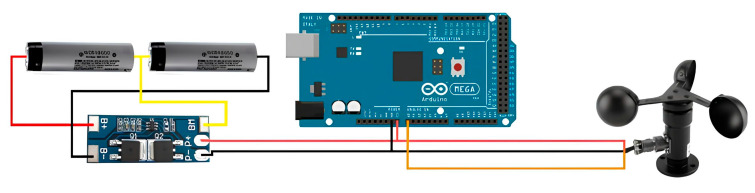
Connection diagram of the wind speed sensor and the protection board of the 18650 lithium-ion battery.

**Figure 6 sensors-23-08645-f006:**
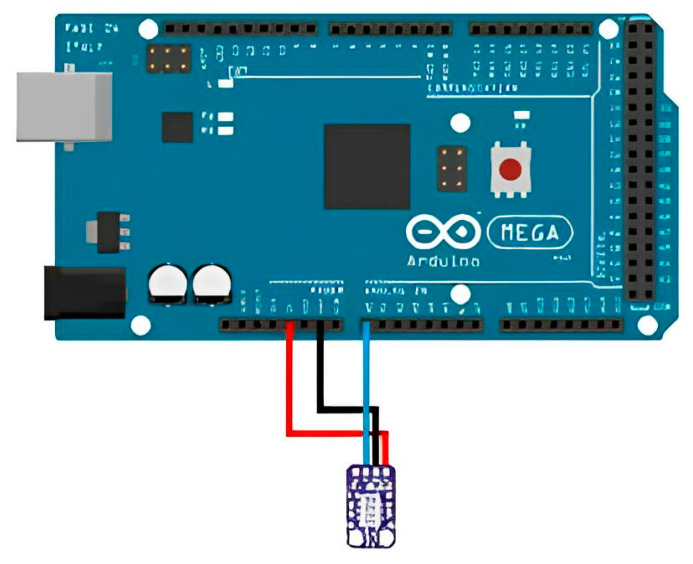
Eu-M-CH4-OD methane sensor connection diagram.

**Figure 7 sensors-23-08645-f007:**
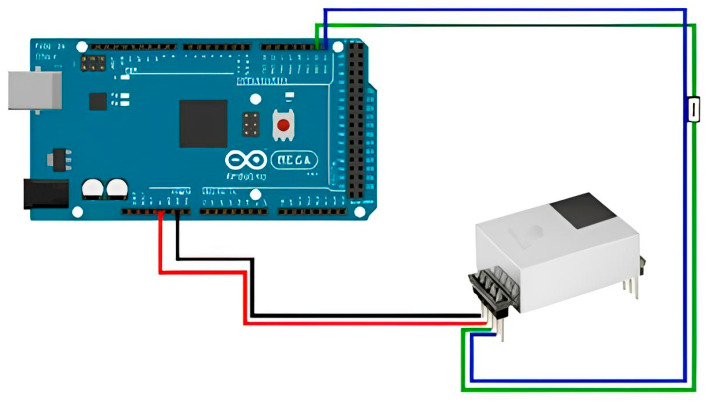
Connection diagram for carbon dioxide sensor CM1107N.

**Figure 8 sensors-23-08645-f008:**
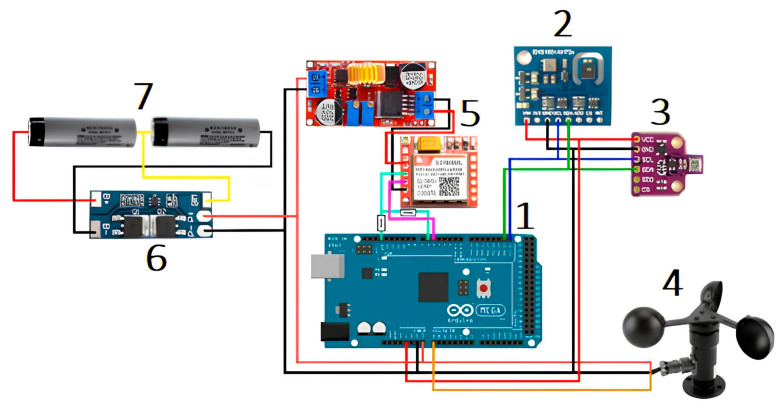
Wiring diagram of the device prototype.

**Figure 9 sensors-23-08645-f009:**
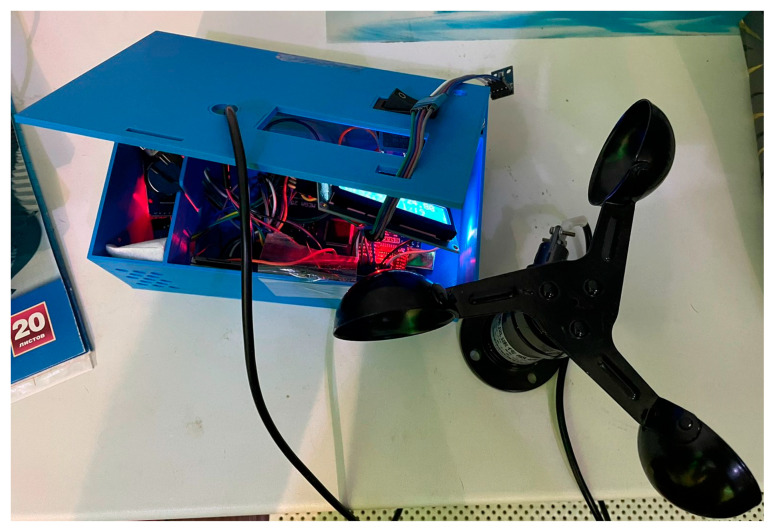
Photo of the control device.

**Figure 10 sensors-23-08645-f010:**
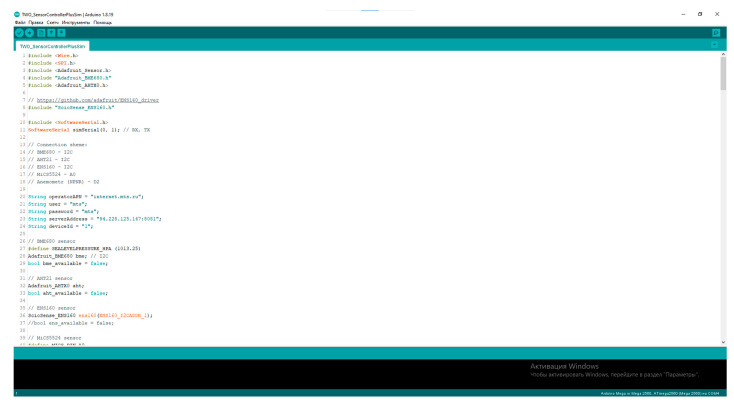
Variable editor window.

**Figure 11 sensors-23-08645-f011:**
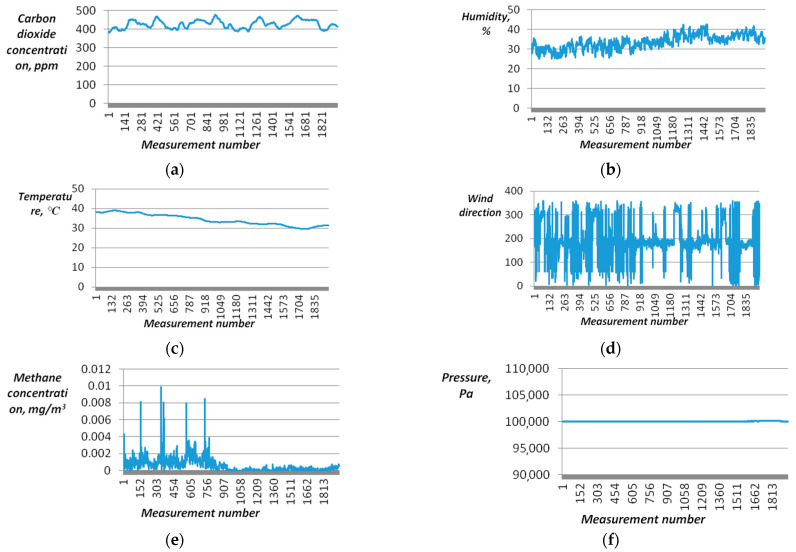
Time series graphs: (**a**)—concentrations of carbon dioxide; (**b**)—humidity of atmospheric air; (**c**)—ambient air temperature; (**d**)—direction of vertical air flow; (**e**)—methane concentrations; (**f**)—atmospheric air pressure.

**Table 1 sensors-23-08645-t001:** Comparative analysis of LiCor open- and closed-type gas analyzers.

Characteristic Name	LI-7500DS	LI-7200RS
Type of gas analyzer	non-dispersive infrared open gas analyzer	non-dispersive infrared spectroscopy
Operating temperature range	from −25 to 50 °C	from −25 to +50 °C
Range of measured concentrations (CO_2_)	from 0 to 3000 ppm mol	from 0 to 3000 ppm mol
Range of measured concentrations (H_2_O)	from 0 to 60 mmol/mol	from 0 to 60 mmol/mol
Uncertainty for CO_2_	1%	In the range of 1%
Uncertainty for H_2_O	1%	In the range of 2%
Resolution (RMS at 370 ppm CO_2_ and 10 ppm mol H_2_O) for CO_2_	at 5 hertz—0.08 ppm; at 10 hertz—0.11 ppm; at 20 hertz—0.16 ppm	at 5 hertz—0.08 ppm; at 10 hertz—0.11 ppm; at 20 hertz—0.16 ppm
Drift (% change relative to °C) for CO_2_	±0.02% for standard, ±0.1% for maximum value	±0.02% (typical)/±0.1% (maximum)
Drift (%change relative to °C) for H_2_O	±0.15% for standard, ±0.30% for maximum value	±0.15% (typical)/±0.30% (maximum)
Sensitivity of CO_2_ to H_2_O (mol CO_2_/mol H_2_O)	±2.00 × 10^−5^ for standard, ±4.00 × 10^−5^ for maximum value	±2.00 × 10^−5^ (typical)/±4.00 × 10^−5^ (maximum)
Sensitivity of H_2_O to CO_2_ (mol H_2_O/mol CO_2_)	±0.02 for standard, ±0.05 for maximum value	±0.02 (typical)/±0.05 (maximum)
Power supply	10.5–30 volt DC	10.5–30 volt DC

**Table 2 sensors-23-08645-t002:** Technical characteristics of the Picarro G2508 gas analyzer.

Specification	N_2_O	CH_4_	CO_2_	NH_3_	H_2_O
Accuracy (1σ)	<25 ppb + 0.05% of reading Typical = 5.0 ppb	<10 ppb + 0.05% of reading Typical = 0.3 ppb	<600 ppb + 0.05% of reading Typical = 240 ppb	<5 ppb + 0.05% of reading Typical = 0.16 ppb	<500 ppm
Accuracy 1 min (1σ)	<10 ppb + 0.05% of reading Typical = 1.1 ppb *	<7 ppb + 0.05% of reading Typical = 0.1 ppb *	<300 ppb + 0.05% of reading Typical = 74 ppb *	<3 ppb + 0.05% of reading Typical = 0.07 ppb *	<250 ppm
Measurement range	0–400 ppm	0–15 ppm	0.02–2%	0–2 ppm	0–7%
Response time	<8 s	<8 s	<8 s	<8 s	<8 s

**Table 3 sensors-23-08645-t003:** Comparative analysis of the temperature sensors.

Sensor	CJMCU-811	ENS160 + AHT21	GY-SGP30
Appearance	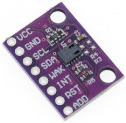	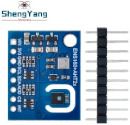	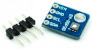
Interface	I2C	I2C и SPI	I2C
Supply voltage, V	1.8…3.3	1.7–2	1.8…5
Consumed current, mA	Up to 30	10–85	48 mA in measurement modeup to 10 nA in sleep mode
Dimensions, mm	14 × 20	19.2 × 24	13 × 10.5 × 2.6
Ambient temperature range, °C	−40 to +85	−40 to +85	−40 to +85
Ambient humidity range, %RH	from 5 to 95	from 5 to 95	from 5 to 95

**Table 4 sensors-23-08645-t004:** Comparative analysis of the sensors for measuring humidity and pressure.

Pressure Sensors	BMP280	LPS22HB	BME680
Appearance	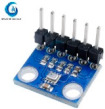	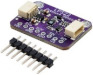	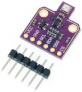
Interface	SPI/I2C	SPI/IIC	SPI/I2C
Operating voltage, V	3.3–5	1.7–3.6 B	1.7–3.6
Pressure range, hPa	300–1100	260–1260	300–1100
Relative accuracy, hPa	± 12	-	-
Temperature range, °C	−40 to + 85	−40 to + 125	−40 to + 85
Humidity, %	100	0–95	0–100

**Table 5 sensors-23-08645-t005:** Comparison table of the carbon dioxide sensors.

Parameters	MH-Z14A NDIR CO_2_ Sensor	IRC-A1	CM1107N	MSH-DS/HC/CO_2_	INP-20-CO_2_ T-NC
Sensor type	optical	optical	optical	optical	optical
Measurement range	0–10,000 ppm	0–5000 ppm 0–5% volume 0–20% volume 0–100% volume	0–5000 ppm	0–2% volume	0–20% volume
Interface	UART	UART	UART	UART	UART
Response time, s	120	40	30	30	30
Working conditions	0–50 °C	−20–55 °C	−10–50 °C	−20–50 °C	5–50 °C
Error	±(50 ppm + 3% reading value)	±1…4%	±(30 ppm + 3% of reading)	±2%	±1% of reading value
Manufacturer	WINSEN, PRC	Alphasense, UK	Cubic, PRC	Dynament, UK	NET, Italy

**Table 6 sensors-23-08645-t006:** Comparative analysis of the methane sensors.

Parameters	MSH-P/HCP	Eu-M-CH4-OD	F3-042107-05000	Gasboard-2500
Sensor type	optical	optical	optical	optical
Measurement range	0–5%, 0–100% volume	0–2.5% oб.	0–10% oб.	0–5% V
Interface	UART	UART	RTU	UART
Response time, sec	30	30	1	20
Working conditions	−20–50 °C	−40–50 °C	0–50 °C	−25–55 °C
Error	±2%	±1%	0.001 Vol.-%	±1%
Manufacturer	Dynament, UK	InformAnalytics, Russia	Smartgas, Germany	Cubic, PRC

**Table 7 sensors-23-08645-t007:** Technical characteristics of the anemometers used for measuring wind speed.

Anemometer	SM5386V	BGT-FS1 RS485	RK100-02
Appearance	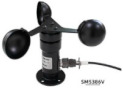	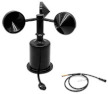	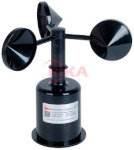
Output mode	RS485/0–5 V/0–10 V	0~5 V; 1~5 V; 0~2.5 V	5 V, 12−24 V
Starting wind speed, m/s	0.1	≤0.5 m/s	<0.5 m/s
Range, m/s	0–30	0~45/0~70	0~45
Error, m/s	± 0.5	± (0.3 + 0.03 V)	±(0.3 + 0.03 V)
Working environment humidity RH, %	0–95	≤100	20–90
Operating temperature, °C	−40~85	−35~60	−40~50
Power, W	1/4

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
