# Peer review of "Technical Solution for Monitoring Climatically Active Gases Using the Turbulent Pulsation Method"

_sensors, 2023, doi:10.3390/s23208645_

Round 1
Reviewer 1 Report
The manuscript deals with a technical solution for monitoring climatically active gases by the turbulent pulsation method. Sensors for greenhouse gases, temperature, relative humidity, pressure and wind intensity/velocity are defined for measurements including electronics. The study is interesting. However, some Revisions are necessary:
1. Materials and Methods section needs to be improved: additional technical description of sensors and analyzers are useful. As an example, a shift of Table 2 from Introduction to Materials and Methods is suggested. Also, datasheests of sensors need to be added as well. Finally, description of electronics should be beneficial shifted from Results to Materials and Methods.
2. Some commercial sensors for GHG are cited mainly from China and Russia. However, additional available commercial sensors from Europe, Japan and USA should be cited in the Table 4 as well.
3. Results should be improved by time-series of GHG sensors, temperature, relative humidity, pressure, wind intensity/speed versus time as measured by used sensors. This is not present in the Results section.
4. Literature should be added by recent papers on topic.
Major Revisions are suggested before publication.
Moderate revision of English should be beneficial
Author Response
Dear reviewer.
We thank you for your valuable recommendations. We have made edits to the text of the article. We have made corrections in the article. They are highlighted in red. Please find our replies as follow:
Reviewer 1:
The manuscript deals with a technical solution for monitoring climatically active gases by the turbulent pulsation method. Sensors for greenhouse gases, temperature, relative humidity, pressure and wind intensity/velocity are defined for measurements including electronics. The study is interesting. However, some Revisions are necessary:
1. Materials and Methods section needs to be improved: additional technical description of sensors and analyzers are useful. As an example, a shift of Table 2 from Introduction to Materials and Methods is suggested. Also, datasheests of sensors need to be added as well. Finally, description of electronics should be beneficial shifted from Results to Materials and Methods. —
Sensors have been added to the tables. The description of sensors has been transferred to Materials and Methods.
2. Some commercial sensors for GHG are cited mainly from China and Russia. However, additional available commercial sensors from Europe, Japan and USA should be cited in the Table 4 as well. —
Sensors from European countries have been added to Table 5
3. Results should be improved by time-series of GHG sensors, temperature, relative humidity, pressure, wind intensity/speed versus time as measured by used sensors. This is not present in the Results section. —
Added photos of the control device itself. Added measurement results of the device: data arrays, graphs are given
4. Literature should be added by recent papers on topic. —
Added 7 references to literature sources and their description in Introduction
Reviewer 2 Report
The turbulent pulsation method, which is announced in the title has not been presented in the manuscript. Authors described the multi sesnor measurement set-up, how they chose the sensors and how the sensors were connected to the ARDIUNO board. The manuscript prsents the description of the activities requiring some technical and enginnering skills, but it is not clear what has been achieved that is new, or in any way progressive with respect to the current state of art. It has not been explained how the system realises the turbulent pulsation method. The measurement system operation has not been deminstrated. Therefore, the work is percieved as incomplete.
Additional comments.
The objectove of the introduction is unclear. It seems to be a set of looslely connected bits and peices of information.
The section Materials and Methods has a residual charatcer. It is quite unusual.
Author Response
Dear reviewer.
We thank you for your valuable recommendations. We have made edits to the text of the article. We have made corrections in the article. They are highlighted in red. Please find our replies as below:
The turbulent pulsation method, which is announced in the title has not been presented in the manuscript. Authors described the multi sensor measurement set-up, how they chose the sensors and how the sensors were connected to the ARDIUNO board. The manuscript presents the description of the activities requiring some technical and engineering skills, but it is not clear what has been achieved that is new, or in any way progressive with respect to the current state of art. It has not been explained how the system realises the turbulent pulsation method. The measurement system operation has not been demonstrated. Therefore, the work is perceived as incomplete.
Additional comments.
The objective of the introduction is unclear. It seems to be a set of loosely connected bits and pieces of information. —
Reply: The purpose of the work has been changed. The need for work is described in more detail
The section Materials and Methods has a residual character. It is quite unusual. —
Reply: Materials and methods have been rewritten
Reviewer 3 Report
The manuscript requires thorough revision to enhance the coherence between its contents.
I recommend seeking the assistance of a native English speaker with expertise in the field to diligently revise the language of the manuscript.
Author Response
Dear reviewer.
We thank you for your valuable recommendations. We have made edits to the text of the article. The manuscript is proofread and corrected by a native speaker. We have made corrections in the article. They are highlighted in red.
Round 2
Reviewer 1 Report
The revised manuscript is improved including reviewer inputs. Now, it is publishable in the present form.
Minor editing of English is suggested.
Reviewer 3 Report
The manuscript introduces an innovative technical solution for investigating the movement of gases within atmospheric air by harnessing turbulent pulsations. Additionally, it presents a well-developed multicomponent control system designed for measuring and recording various atmospheric characteristics. This research holds significant promise in advancing the field of comprehensive investigations into the dynamics of climatically active gases. I recommend accepting the current version of this manuscript.